# Molecular Clustering of Metabolic Dysfunction-Associated Steatotic Liver Disease Based on Transcriptome Analysis

**DOI:** 10.3390/diagnostics15030342

**Published:** 2025-01-31

**Authors:** Gina Ryu, Eileen Laurel Yoon, Wankyu Kim, Dae Won Jun

**Affiliations:** 1Department of Life Sciences, College of Natural Science, Ewha Womans University, Seoul 03760, Republic of Korea; rimeless@ewha.ac.kr; 2Department of Internal Medicine, College of Medicine, Hanyang University, Seoul 04763, Republic of Korea; mseileen80@hanyang.ac.kr; 3Hanyang Institute of Bioscience and Biotechnology, Hanyang University, Seoul 04763, Republic of Korea

**Keywords:** MASLD, phenotype, cluster, molecular

## Abstract

**Background:** Metabolic dysfunction-associated steatotic liver disease (MASLD) is a complex metabolic disorder with a diverse spectrum. This study aimed to classify patients with MASLD into molecular subtypes based on the underlying pathophysiology. **Methods:** We performed high-throughput RNA sequencing on 164 liver tissue samples from healthy controls and patients with MASLD. The clustering was based on individual genes or pathways that showed high variation across the samples. Second, the clustering was based on single-sample gene set enrichment analysis. **Results:** Optimal homogeneity was achieved by dividing the samples into four clusters (one healthy control and three MASLD clusters I–III) based on the top genes or pathways with differences across the samples. No significant differences were observed in waist circumference, blood pressure, glucose, alanine transaminase (ALT), or aspartate transferase (AST) levels between cluster I patients with MASLD and the healthy controls. Cluster I showed the clinical features of lean MASLD. Cluster III of MASLD demonstrated hypertension and a T2DM prevalence of 57.1% and 50.0%, respectively, with a significantly higher fibrosis burden (stage of fibrosis, liver stiffness, and FIB-4 value) than clusters I and II. Cluster III was associated with severe fibrosis and abnormal glucose homeostasis. In MASLD cluster I, the sphingolipid and GPCR pathways were upregulated, whereas the complement and phagocytosis pathways were downregulated. In MASLD cluster II, the cell cycle and NOTCH3 pathways increased, whereas the PI3K and insulin-related pathways decreased. In MASLD cluster III, increased activity occurred in the interleukin-2 and -4 and extracellular matrix pathways, coupled with decreased activity in the serotonin 2A and B pathways. **Conclusions:** MASLD can be divided into three distinct molecular phenotypes, wherein each is characterized by a specific molecular pathway.

## 1. Introduction

Metabolic dysfunction-associated steatotic liver disease (MASLD) affects approximately 25% of the global population and is the leading cause of hepatocellular carcinoma and liver transplantation [1,2,3]. The clinical courses and the characteristics of MASLD vary widely among individuals. It reflects that MASLD is a multifactorial disease, including genetic predisposition, dietary habits, physical inactivity, intestinal dysbiosis, sarcopenia, and metabolic dysfunction [4,5]. Understanding the heterogeneity and pathophysiology of MASLD is critical for developing personalized therapeutic approaches.

However, the practical application of personalized treatment strategies based on these insights remains limited. Although almost all physicians recognize the heterogeneous pathophysiology of MASLD, an effective model for classifying patients and applying personalized treatments has yet to be established. Yi et al. performed the clinical phenotyping of MASLD using 21 clinical parameters from the US National Health and NHANES data [6]. The patients with MASLD were divided into three clusters. Cluster I was characterized by young female patients with relatively healthy metabolic profiles. Obese women with significant insulin resistance, diabetes, and advanced fibrosis characterized cluster II. Lastly, male patients with hypertension, atherogenic dyslipidemia, and liver and kidney damage characterized cluster III. Another study by Ye et al. reported similar results. This phenotyping study, which utilized clinical parameters from the UK Biobank and a Chinese cohort, classified patients into five clusters: cluster 1 (obesity), cluster 2 (age-related), cluster 3 (severe insulin resistance-related), cluster 4 (dyslipidemia), and cluster 5 (severe mixed hyperlipidemia) [7]. This approach of MASLD clustering based on clinical phenotyping offers the advantage of classifying patients with MASLD more intuitively based on clinical indicators. However, information on the treatment strategies and pathomechanisms of the individual clusters is limited. Clustering MASLD based on clinical data and underlying pathophysiology is expected to enable the development of customized treatment strategies. To enhance the therapeutic outcomes of patients with MASLD, classifying them based on the underlying mechanisms that drive the disease development is crucial. Customized treatment strategies tailored to the specific pathogenesis of individual MASLD cases are anticipated to increase the treatment success rate and accelerate the development of MASLD-targeted pharmacotherapies.

This study aimed to classify individuals with MASLD into several subtypes based on the unsupervised clustering of transcriptome data, which further elucidated the different signatures of the genes and the molecular pathways.

## 2. Materials and Methods

### 2.1. Study Design

This prospective study used the data collected from a tertiary hospital. Informed consent was obtained from all study participants, and the study protocol was approved by the Institutional Review Board (HY-IRB 2019-12-028-019).

### 2.2. Inclusion and Exclusion Criteria

This study included 164 liver tissue samples from healthy individuals and patients with MASLD. Eligible participants were adults aged over 18 undergoing prophylactic cholecystectomy. Steatotic liver disease (SLD) is diagnosed based on liver histology. Moreover, MASLD was defined as the presence of histological hepatic steatosis, no significant alcohol consumption, and at least one of the five cardiometabolic risk factors [8]. The exclusion criteria were as follows: (1) having concurrent acute cholecystitis, (2) the presence of alanine aminotransferase (ALT) exceeding five times the upper limit of the normal range, (3) having any other chronic liver disease than MASLD encompassing viral hepatitis (e.g., positive results for the hepatitis B surface antigen or hepatitis C virus antibody) and alcohol-related liver disease (significant alcohol consumption of ≥ 210 g/week for men and ≥ 140 g/week for women).

### 2.3. Histological Interpretation

Liver biopsy specimens were used for the histological diagnosis of all subjects. Using standard procedures, liver samples were embedded in paraffin blocks and stained with hematoxylin and eosin (H&E) and Masson’s trichrome. An experienced liver pathologist, using the NASH Clinical Research Network histological scoring system, centrally evaluated all liver biopsies at one time for this study. The Kleiner classification was used to score liver fibrosis on a 5-point scale (0–4), with scores of 3-4 indicating advanced fibrosis.

### 2.4. Bulk RNA Sequencing

The samples for RNA extraction were titrated using a TissueLyzer (Qiagen, Hilden, Germany). A cDNA library was constructed using 500 ng of RNA per sample. The RNA was fragmented, double-stranded cDNA (dsDNA) was synthesized, and adapter sequences were linked by combining thymine residues at the 3′-end with adenine residues on the fragmented DNA. The library was amplified by PCR, and the byproducts were removed using magnetic beads. The amplified DNA was single-stranded and immobilized on a flow cell. Adapter sequences and complementary primers were combined, and sequencing was performed over 200 cycles, measuring the light emitted by fluorescently labeled bases (A, T, G, and C). The RNA sequencing reads were trimmed using fastp-0.23.4, aligned with STAR v2.7.1a, and raw read counts were generated using HTSeq-2.0.3.

### 2.5. MASLD Clustering Using Transcriptome Data

Clustering within the MASLD cohort was conducted using non-negative matrix factorization (NMF). The clustering process was unsupervised, utilizing the top 10–40% of genes or pathways with high variation, identified using the median absolute deviation (MAD). The number of clusters was determined using the cophenetic correlation coefficient, which measures cluster stability. The optimal number of clusters was selected at the elbow point of the cophenetic correlation plot, balancing robust cluster detection with practical interpretability.

### 2.6. Single-Sample Gene Set Variation Analysis (ssGSVA)

ssGSVA computes enrichment scores for individual sample–gene set combinations, revealing how genes within a set are regulated in a sample [9]. Unlike conventional GSEA, which assesses gene sets across samples, ssGSVA offers a sample-centric perspective by converting gene expression profiles into pathway activity levels. Enrichment scores for the pathways from the MsigDB C2 collection were calculated for each sample [10].

### 2.7. Validation Using Publicly Available MASLD Datasets

The public dataset GSE135251 of patients was obtained from the Gene Expression Omnibus (GEO) [11]. It includes transcriptome profiles from 206 MASLD patients categorized at various stages of hepatic fibrosis. We clustered 206 samples using gene/pathway signatures by applying single-sample gene set variation analysis (ssGSVA). The ssGSVA scores were normalized across all samples, and each sample was classified into one of four clusters.

### 2.8. Identification of Cluster-Specific Cell Type

Liver cell-type markers were accessed from the MsigDB C8 collection, which is based on a human liver cell atlas [12]. Cluster-specific signatures were identified for each MASLD cluster, where the mean expression value was significantly higher compared with all other clusters.

### 2.9. In Silico Prediction of Potential Targets Using Drug-Induced Transcriptome

Target gene analysis for each MASLD cluster was conducted using a connectivity map (CMAP) dataset to identify potential therapeutic targets [13]. The known targets for the CMAP drugs were compiled from public databases of drug–gene interactions [14]. First, the drugs with a strong inverse expression pattern (top 10%; 657 drug signatures of HEPG2 and HUH7 liver cells) were identified for each cluster by comparing up- and down-signatures between the MASLD cluster and CMAP drugs, respectively. Second, the consensus targets of these drugs were determined by calculating the enrichment factor (EF) of each target. The EF was measured by calculating the overlap score ratio of the actual and expected values.
$$Enrichment\;Factor\left(\mathrm{set}\;A,\;\mathrm{set}\;B\right)=\frac{observed\;counts+1}{expected\;counts+1}=\frac{N_{A\cap B}+1}{\frac{N_{A}\times N_{B}}{N_{total}}+1}$$

### 2.10. Statistical Analysis

Differences in clinical parameters between the healthy control group and patients with MASLD were assessed using ANOVA for continuous variables and the chi-square test for categorical variables. The Kruskal–Wallis H test was used to analyze the differences in gene expression between the normal control group and patients with MASLD. Transcriptome analysis was performed using R 4.1.3, with clustering conducted using the NMF_0.26 package. Single-sample GSVA was carried out with the GSVA package, and pathway enrichment analysis was performed by calculating enrichment scores and significance using a hypergeometric test. Heatmaps and plots were visualized using the ComplexHeatmap_2.10.0 or ggplot2_3.5.1 packages.

## 3. Results

### 3.1. Baseline Characteristics

The clinical characteristics of the 102 patients with MASLD and 62 non-SLD controls are summarized in Appendix A. Among the patients with MASLD, 76 were diagnosed with MASLD without metabolic dysfunction-associated steatohepatitis (MASH), and 26 were diagnosed with MASH. Compared with the non-SLD control group, patients with MASLD exhibited a higher prevalence of diabetes and hypertension, along with elevated levels of body mass index, triglycerides, fasting glucose, alanine aminotransferase, aspartate aminotransferase (AST), and hemoglobin A1c.

### 3.2. Clustering of MASLD According to Gene-Wise and Pathway-Wise Approaches

We independently clustered patients with MASLD in two ways: gene- and pathway-wise clustering, as depicted in Figure 1. We independently performed NMF clustering based on both (i) gene expression and (ii) pathways from a single sample (Figure 1A). Maximum homogeneity was achieved among the four clusters by utilizing the top 10% of genes and the top 30% of pathways. The optimal number of clusters was set to four because the cophenetic correlation plots consistently indicated the highest cluster stability while providing sufficient diversity for interpretation in the gene- (Figure 1B) and pathway-level clustering results (Figure 1C). Clustering into one normal and three MASLD clusters (indicated as clusters I, II, and III among MASLD) maintained a cophenetic correlation coefficient of 0.96~0.99, indicating high homogeneity within each cluster (Figure 1D,E).

### 3.3. Molecular Pathways According to MASLD Clusters Using Signature Genes and Single-Sample Pathway-Based Clustering

Our study identified 1211 upregulated and 1062 downregulated signature genes specific to each cluster in the control and MASLD groups, as detailed in Figure 2A,B. Cluster-specific molecular pathways were extracted for each MASLD cluster (clusters I to III) by analyzing the significantly upregulated and downregulated genes. Using this gene-wise clustering approach, we identified 140 distinct pathways (Figure 2C). Additionally, we conducted pathway-wise clustering of individual samples independent of the gene-wise approach. This method provided a clearer distinction between clusters with less noise than the gene-wise approach, as depicted in Figure 2D–F. This analysis also revealed the 140 distinct molecular pathways characteristic of each cluster.

### 3.4. Comparison of Gene- and Pathway-Based Clustering of MASLD

We identified 140 pathways across different clusters. Notably, the two clustering approaches showed a high level of similarity, with an agreement rate of 84.9% (90/106) for the same patient set (Figure 3A). The characteristics of major pathways in each MASLD cluster are shown in Figure 3B. Cluster I exhibited an increase in the sphingolipid and GPCR pathways, particularly the olfactory receptors and the thyroid hormone pathway. Conversely, a decrease was observed in the complement pathway levels. Cluster II demonstrated elevated SIRT1 and NOTCH3 pathway activity, with reduced PI3K and insulin-related pathway activity. Cluster III showed increased interleukin-2 and -4 and fibrosis-related pathways, including the extracellular matrix (ECM) and folate pathways. Moreover, a decrease was found in serotonin 2A and B levels as well as cell cycle-related pathways. This clustering provided a nuanced view of the biochemical pathways potentially involved in MASLD pathogenesis, guiding future therapeutic interventions.

### 3.5. External Validation of Molecular Phenotype of MASLD

We further sought to validate the reproducibility of our clustering and signature identification methods using an independent dataset of patients with MASLD. Using the expression profiles from 216 patients in the GSE135251 dataset, we performed both gene- and pathway-wise clustering, resulting in the stratification of patients into three distinct groups (Figure 4). Molecular-based MASLD clustering showed strong concordance with histological severity indicators, such as stage, NAS score, and fibrosis, as well as with clinical parameters, ranking in the order of clusters III > II > I. Specifically, MASLD cluster I was characterized by increased GPCR pathways, notably olfactory receptors. Cluster II exhibited reduced activity in the PI3K, FGFR, and insulin-related pathways, indicating the downregulation of signaling pathways associated with cell growth and metabolism. Cluster III was distinguished by the pronounced enhancement of fibrosis and the upregulation of interleukin-related pathways, along with a reduction in cell cycle progression and zinc transport pathways.

### 3.6. Clinical Characteristics According to Molecular Classifications of MASLD

The clinical characteristics of the three MASLD clusters are shown in Appendix A. MASLD cluster I had a similar prevalence of T2DM and hypertension to healthy controls. Additionally, no significant differences were observed in waist circumference, fasting blood glucose, ALT, and AST levels compared with the normal controls. Cluster I exhibited clinical characteristics of lean patients with MASLD, with a relatively mild degree of metabolic dysfunction. In contrast, MASLD cluster III had a high prevalence of hypertension (57.1%) and diabetes (50%), as well as advanced stages of intrahepatic fibrosis, as seen in VCTE and liver biopsy. Cluster III demonstrated the clinical characteristics of abnormal insulin resistance and significant hepatic fibrosis, fitting the profile of diabetic fibrotic MASLD. Cluster II displayed the clinical characteristics of obese patients with MASLD, with a high frequency of metabolic dysfunction, which is most commonly observed in clinical practice.

### 3.7. Molecular Signal Pathway According to Type of Liver Cell

Although bulk RNA transcriptomes cannot classify individual cell types (e.g., hepatocytes, stellate cells, and Kupffer cells) in the liver, we analyzed gene and pathway expression in major liver cells according to clusters. Interestingly, the signature genes and pathways were differentially expressed across clusters, even within the same cell type (Figure 5). In hepatocyte-like cells, cluster I hepatocytes exhibited upregulation in the amino acid and folate biosynthesis pathways, alongside downregulation in the semaphorin, IL-8, and cell–matrix adhesion pathways. Cluster II hepatocytes demonstrated enhanced retinol metabolism and peroxisomal functions while showing decreased activity in the FOXO, PPAR, and glucagon signaling pathways. Cluster III hepatocytes were characterized by the upregulation of ANGPTL8, leptin, and adipocytokine signaling, with the concurrent downregulation of pentose and glucuronate interconversion, as well as the terpenoid metabolism and steroid hormone biosynthesis pathways. In Kupffer-like cells, cluster I Kupffer cells were enriched in pathways related to cytokine signaling, leukotrienes, and interleukins. Cluster II Kupffer cells were characterized by antigen presentation and phagosome pathways. Cluster III Kupffer cells exhibited an increase in complement pathways. Among the stellate cell marker genes, the normal cluster was enriched in integrin. Inflammation and fibrosis-related pathways, such as TGF-beta receptor signaling and connective tissue development, characterized cluster III.

### 3.8. Potential Therapeutic Targets for the Molecular Phenotype of MASLD

Given the heterogeneity of MASLD, it is necessary to consider therapeutic targets tailored to each subtype. We conducted an in silico inference of potential targets based on a large-scale drug-induced transcriptome dataset, a connectivity map [15]. The approach relies on inverse pattern matching between disease- and drug-related expression signatures, assuming that such drugs may drive cells from a disease state close to normal. Accordingly, the consensus or enriched targets of these drugs were considered as potential targets for treatment (Figure 6). Consequently, we identified a list of potential targets based on expression signature genes for MASLD clusters I–III that are strongly related to MASLD, including those undergoing clinical trials. In cluster I, well-known targets, such as BRD2 and BRD3, appeared significant, with an EF 3.5 of 3.6 [16]. In cluster II, PRKAG2 is known to exacerbate hepatic inflammation by modulating TNF-alpha [17]. MAP4K also stood out as a pivotal regulator of hepatocyte regeneration processes, suggesting that inhibition of MKP4K is an effective approach to liver repair and regeneration [18]. Repression of LRRK2, a regulator of lipid metabolism, is reported to promote lipid catabolism and suppress inflammation in MASLD models [19]. The PI3K pathway is a down-signature pathway of cluster II and is also predicted as a potential target. In cluster III, MTR, a methionine synthase, is responsible for homocysteine metabolism and is the top candidate for EF 5.1, which may act as a mitigating factor for oxidative stress and inflammation by elevating homocysteine [20]. CCR5 is known to recruit and maturate inflammatory macrophages [21] and its targeted drug, leronlimab, is under clinical trials. Resmetirom, a THRB agonist, has recently been approved as the first MASLD drug by the FDA [22]. HMGCR is the target of simvastatin, a drug subjected to clinical trials demonstrating its efficacy in reducing hepatic cholesterol synthesis [23]. RAP1A plays a role in liver inflammation and fibrosis, and the activation of RAP1A could protect against fatty liver and MASLD development [24]. A non-mitogenic variant of FGF1 reduces liver weight, lipid deposition, and inflammation in mouse models of MASLD [25]. DPP4 is a well-established MASLD treatment target [26]. DPP4 inhibitors, such as sitagliptin and gemigliptin, have shown potential in ameliorating hepatic steatosis and inflammation in MASLD by modulating glucose metabolism and inflammatory pathways [27].

## 4. Discussions

MASLD displays highly heterogeneous clinical characteristics and courses influenced by various metabolic, genetic, and dietary factors [28]. This indicates significant variability in the pathophysiology of patients with the same clinical phenotype [2]. In this study, we applied NMF clustering to obtain an unbiased and comprehensive screening of expression markers, resulting in three distinct molecular subtypes (clusters). Cluster I showed activation of the sphingolipid pathway and decreased complement pathway activity. The dysregulation of sphingolipids contributes to the development of steatosis and is also linked to inflammation and cirrhosis [29]. GPCRs, which function as receptors for bile acids and free fatty acids, have gained attention as potential therapeutic targets for MASLD [30]. Cluster II showed the activation of cell cycle-related pathways and NOTCH 3, along with decreased PI3K and insulin-related pathways. SIRT1 is closely associated with MASLD pathogenesis due to its role in mitochondrial function restoration and lipid metabolism regulation [31]. The upregulation of NOTCH signaling coupled with the downregulation of the PI3K signaling cascade indicates insulin resistance, as NOTCH signaling is associated with insulin resistance in MASLD [32]. Furthermore, the PI3K pathway is fundamental in mediating insulin’s actions on glucose and lipid metabolism, exacerbating metabolic imbalances [33]. Cluster III displayed the activation of the interleukin-2 and -4 and ECM pathways and reduced serotonin 2A and B and cell cycle-related pathways. The ECM is a well-known driver of fibrosis [34], and a reduction in liver serotonin receptor 2A has been shown to ameliorate hepatic steatosis, indicating its potential role in the progression and treatment of MASLD [35].

Several clinical phenotyping and clustering studies are based on clinical parameters [36]. Yi et al. classified MASLD into three clusters using NHANES data [6]. Cluster 1 was characterized by young female patients with relatively healthy metabolic profiles and a lower prevalence of complications. Obese women with significant insulin resistance, diabetes, inflammation, and advanced fibrosis characterized cluster 2. Cluster 3 was characterized by male patients with hypertension, atherogenic dyslipidemia, and liver and kidney damage. Interestingly, our study performed MASLD clustering based on transcriptome data and showed similar results. In our study, cluster I had a younger age and a higher proportion of females, with better metabolic parameters than clusters II and III (Appendix A). Moreover, cluster III had a higher proportion of males, a higher rate of diabetes, and more fibrosis, showing a pattern similar to the clinical phenotyping suggested by Yi et al. Notably, as observed in the study by Yi et al. and ours, MASLD cluster II exhibited an age-related MASLD pattern with metabolic deterioration that was worse than that of cluster I but better than that of cluster III. A recent study by Ye et al. reported comparable results. Their clinical phenotyping study, which used clinical parameters from the UK Biobank and a Chinese cohort, classified patients into five clusters: cluster 1 (obesity), cluster 2 (age-related), cluster 3 (severe insulin resistance-related), cluster 4 (dyslipidemia), and cluster 5 (severe mixed hyperlipidemia) [7]. Although MASLD clustering based on clinical phenotyping offers the advantage of classifying patients with MASLD more intuitively based on clinical indicators, information on treatment strategies and pathomechanisms for individual clusters is inevitably limited. MASLD molecular clustering based on the transcriptome aligns well with the results of clinical phenotyping. It allows an understanding of the primary pathophysiology of each cluster, suggesting customized treatment strategies based on pathogenesis.

Most transcriptome-related studies on MASLD have focused on identifying novel genes that show significant differences between normal and MASLD tissues to develop novel targets or biomarkers. Govaere et al. characterized the transcriptional changes occurring in liver tissue across the MASLD spectrum as the disease progresses to cirrhosis [11]. They specifically studied significant changes in transcripts at various stages of MASLD according to severity. They found that AKR1B10 and GDF15 were strongly associated with disease activity and the fibrosis stage. Our study focused on pathway-based differences rather than differences in individual gene expression between normal and MASLD samples. Molecular phenotypic analysis of MASLD is anticipated to be instrumental in developing new drugs and establishing customized treatment strategies for individual patients. For example, if a drug targets the FGF pathway, cluster II may be a better treatment target than cluster I, where the FGF pathway is already activated. Conversely, for drugs related to the interleukin pathway, cluster I, in which the interleukin pathway is reduced, may be a more appropriate target than cluster III, in which the interleukin pathway is activated.

Our findings suggest that MASLD can be classified into three clusters based on distinct gene and/or pathway characteristics. Cluster I was characterized by metabolic and signaling alterations, increased sphingolipid metabolism and GPCR activity, which play essential roles in cellular communication, and a decreased immune response. Cluster II reflected enhanced cellular proliferation and repair mechanisms, as indicated by elevated cell cycle activity, and also showed signs of insulin resistance and metabolic dysfunction due to reduced PI3K and insulin signaling. Cluster III was marked by pronounced inflammatory and fibrotic activity, as evidenced by the upregulation of interleukin- and ECM-related pathways. We carried out external validation using an independent public RNA transcriptome dataset (GSE135251, *n* = 206) [11]. Despite differences in disease stage and ethnicity, there were common aspects shared between the two independent clustering results. This suggests that our study captured at least several subtype-specific pathways common to both cohorts. This could provide an interpretable and independent framework compared with conventional clinical features, such as disease stage, metabolic markers, and comorbidities. Furthermore, these results may provide a lead to subtype-specific therapeutics and patient stratifications.

This study had several limitations. First, additional external validation is needed for more diverse MASLD cohorts for generalization. Since each cohort has a different MASLD severity, validation is needed in more diverse cohorts. Although public transcriptome data were also classified into three clusters, and most signature pathways of each cluster overlapped with the derivative cohort, there were certain discrepancies between the two cohorts. There are two main reasons for those discrepancies. The disease severities of the two cohorts were different. The derivation cohort included healthy normal livers, but the validation cohort did not include normal healthy controls. The severity in the validation cohort was more than that in the derivation cohort. The extent of bias and the generalizability of our MASLD clusters will be validated as additional data become available. Second, although this study identified the characteristics of each MASLD cluster based on molecular pathways related to the pathophysiology of fatty liver disease, more practical classification methods and standards are required in clinical practice to effectively classify new patients with MASLD into these three clusters. Third, the primary purpose of MASLD molecular phenotyping is to select customized treatment drugs according to molecular clusters in a diverse spectrum of patients with MASLD. However, this approach requires validation through preclinical and clinical experiments to confirm its efficacy. Nevertheless, the primary purpose of this study was to subclassify the individuals with MASLD based on unsupervised clustering through molecular phenotyping. We presume that the experimental validation of these data should be performed in future studies.

## 5. Conclusions

In conclusion, MASLD can be divided into three molecular phenotypes, each characterized by distinct molecular pathways. Moreover, molecular clustering based on these phenotypes strongly correlated with clinical parameters and histological severity.

## Figures and Tables

**Figure 1 diagnostics-15-00342-f001:**
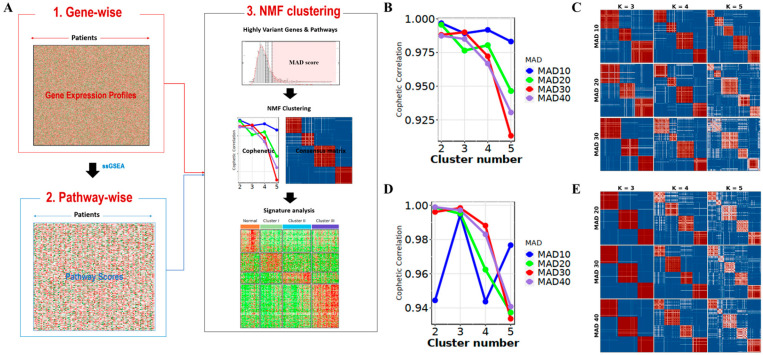
The procedure of NMF clustering and signature selection by gene-wise and pathway-wise approaches. (**A**) The overall procedure of conducting NMF clustering using either gene expression profiles or pathway scores by ssGSVA (single-sample gene set variation analysis). First, the top 10~40% of high-variant genes/pathways were selected according to the MAD (median absolute deviation). Second, NMF clustering was performed, and the optimal number of clusters was determined based on the cophenetic correlation and consensus matrix. Finally, the signature genes/pathways of each cluster were extracted that were significantly up- or downregulated (*t*-test; FDR < 0.01) relative to the other clusters, both individually and collectively. The cophenetic correlation and consensus matrix were based on genes (**B**,**C**) and pathway scores (**D**,**E**). Overall, the cophenetic coefficients consistently formed an elbow at cluster number (k) = 4, with maximal consensus in both approaches.

**Figure 2 diagnostics-15-00342-f002:**
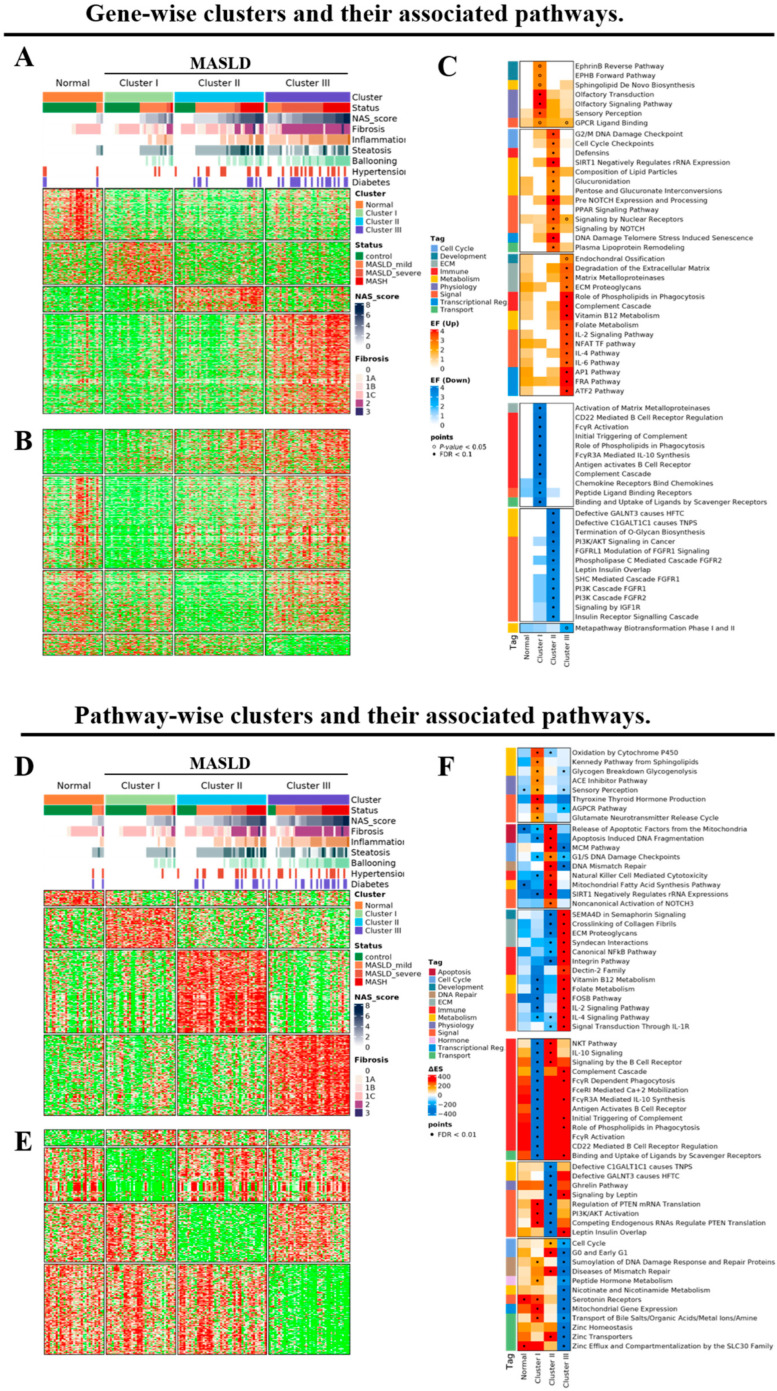
MASLD clustering based on molecular pathways according to gene-wise and pathway-wise approaches. (**A**–**C**) The gene-wise clusters of the normal group and clusters I, II, and III and their associated pathways. (**A**) Up- and (**B**) downregulated relative to the other clusters. (**C**) The selected list of cluster-specific pathways is displayed based on the gene set analysis of both up- and down-signature genes in each cluster compared with the other clusters. (**D**–**F**) The pathway-wise clusters of the normal group and clusters I, II, and III and their associated pathways. (**D**) Up- and (**E**) downregulated relative to the other clusters. (**F**) The selected list of cluster-specific pathways is displayed based on gene set analysis of both up- and down-signature genes in each cluster compared with the other clusters. The color scale is based on the enrichment factor computed for cluster-specific gene sets, and the hypergeometric test measures significance.

**Figure 3 diagnostics-15-00342-f003:**
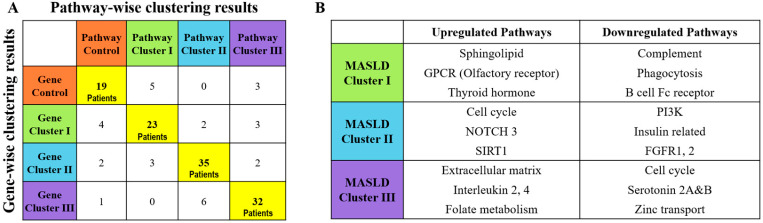
Comparison of gene- and pathway-wise clustering results. (**A**) The comparison of patient classification by the two approaches and (**B**) the representative signature pathways shared by both clustering results.

**Figure 4 diagnostics-15-00342-f004:**
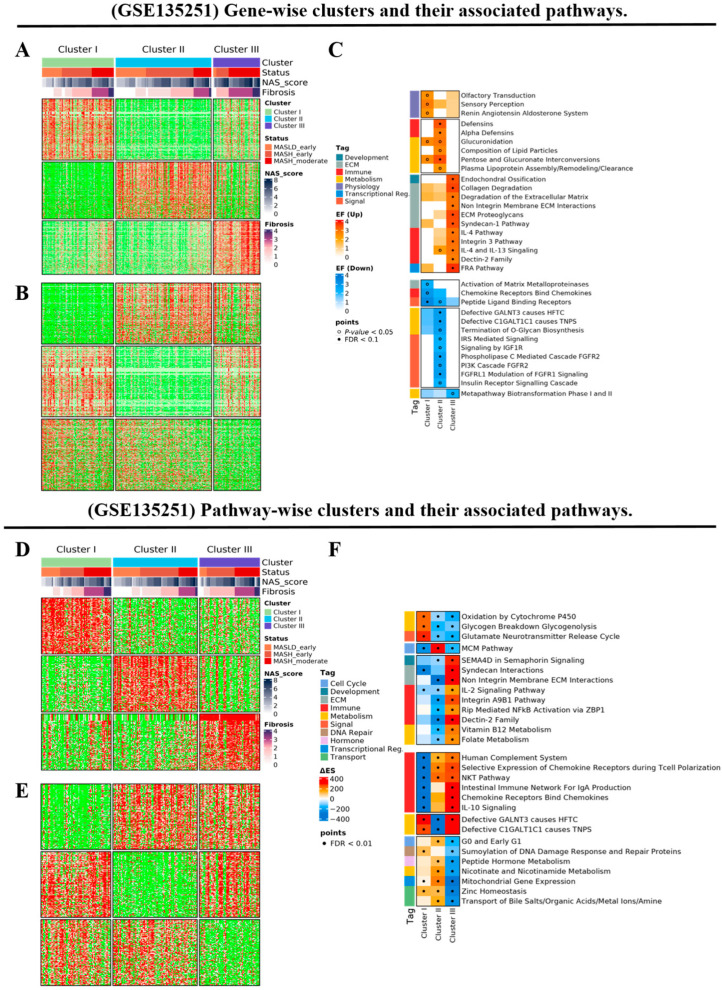
External validation of MASLD clustering based on the molecular pathway. (**A**–**C**) The gene-wise clusters of the normal group and clusters I, II, and III and their associated pathways. (**D**–**F**) The pathway-wise clusters of the normal group and clusters I, II, and III and their associated pathways. (**A**,**D**) Up- and (**B**,**E**) downregulated relative to the other clusters. (**C**,**F**) The selected list of cluster-specific pathways is displayed based on gene set analysis of both up- and down-signature genes in each cluster compared with the other clusters.

**Figure 5 diagnostics-15-00342-f005:**
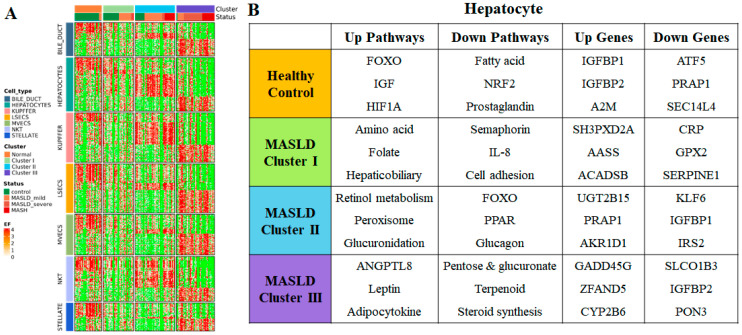
(**A**) The expression of marker genes for major cell types in liver tissue. The markers for the same cell type do not necessarily show a similar pattern across the gene-wise clusters. (**B**) The enriched pathways are displayed for several important cell types in MASLD pathogenesis.

**Figure 6 diagnostics-15-00342-f006:**
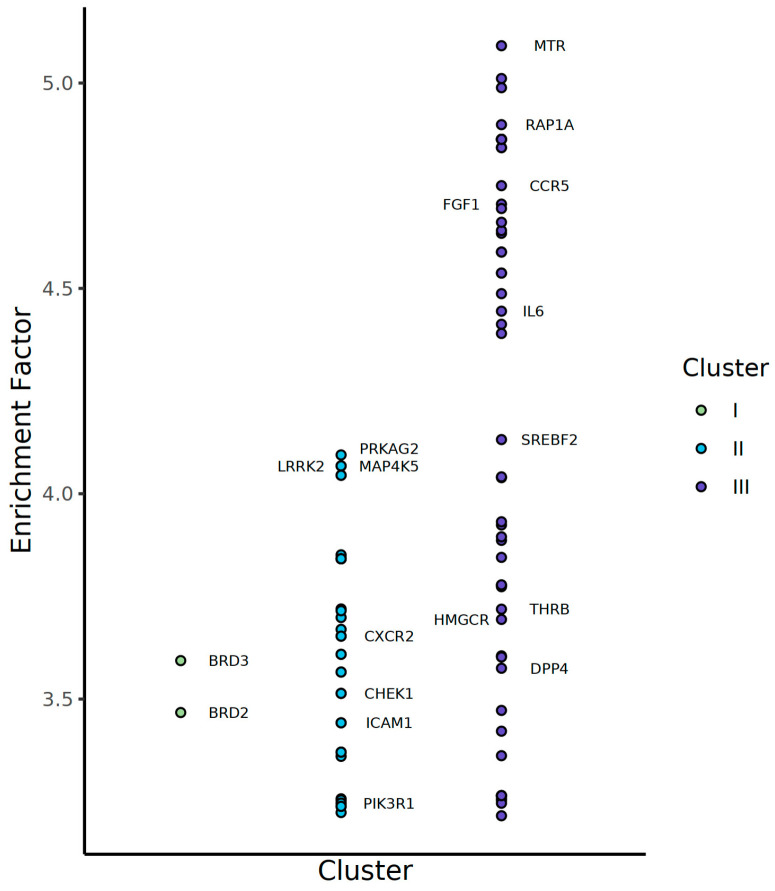
The potential targets of MASLD were inferred using the large-scale drug-induced transcriptome dataset and the up-signatures of each gene-wise cluster. The predicted list includes many known targets for MASLD, including the ones under clinical trials.

## Data Availability

The raw and processed RNA-seq data will be provided upon request upon IRB approval. All other data, such as the external datasets and gene sets, are accessible through the sources mentioned in this manuscript.

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
