# Peer review of "Molecular Clustering of Metabolic Dysfunction-Associated Steatotic Liver Disease Based on Transcriptome Analysis"

_diagnostics, 2025, doi:10.3390/diagnostics15030342_

Round 1

Reviewer 1 Report

Comments and Suggestions for Authors

This study employed transcriptome analysis to classify patients with metabolic dysfunction-associated steatotic liver disease (MASLD) into distinct molecular subtypes, revealing three distinct molecular phenotypes characterized by specific molecular pathways, which strongly correlate with clinical parameters and histological severity. The reviewer has raised several concerns regarding the manuscript's analysis and discussion of molecular clustering in metabolic dysfunction-associated steatotic liver disease (MASLD):

1.    Insufficient Biological Mechanism Discussion: The reviewer notes that the biological mechanisms underlying each molecular subtype are not adequately discussed. Specifically, the role of sphingolipid metabolism and GPCR pathways in Cluster I, the implications of upregulated cell cycle and NOTCH3 pathways in Cluster II, and the significance of inflammatory and fibrosis-related pathways in Cluster III require further elucidation.

2.    Selection of Genes and Pathways: The choice of the top 10%-40% genes and pathways for non-negative matrix factorization (NMF) clustering is questioned. The reviewer asks for the rationale behind this selection and whether it might omit potentially important genes or pathways. They also inquire about the use of alternative parameter ranges or methods, such as principal component analysis (PCA) or hierarchical clustering, and their impact on the results.

3.    Lack of Experimental Validation: The manuscript proposes several candidate drug targets based on transcriptome data but does not discuss their experimental validation in MASLD models or in vitro studies. The reviewer suggests that this lack of validation may diminish the clinical relevance of the proposed targets and recommends incorporating a discussion on existing literature regarding the validation of these targets in MASLD or related diseases.

4.    External Validation Concerns: The use of the GSE135251 dataset for external validation is critiqued due to differences in disease severity between the validation and original cohorts. The reviewer raises concerns about the potential impact of disease staging and sample composition on the applicability of the clustering results and asks if the authors controlled for these variables. They also question the potential for systemic bias in transcriptomic differences due to the datasets being derived from patients of different ethnicities or regions.

5.    Translation to Personalized Treatment: While the article suggests that each cluster may require distinct treatment strategies, it does not discuss how the findings could be translated into personalized therapy options. The reviewer requests a more detailed discussion on the potential therapeutic implications, such as the use of GPCR-targeted drugs for Cluster I, PI3K inhibitors for Cluster II, and antifibrotic or anti-inflammatory treatments for Cluster III.

Reviewer 2 Report

Comments and Suggestions for Authors

This is an interesting study by Ryu et al regarding the use of a tissue biopsy gene expression profile to identify patient clusters of MASLD that could benefit from pathway-specific therapies. The study has significant potential to help researchers in the field identify therapeutic targets. Although the findings are important the study needs to be more analytic in its methodological part. In its current form, the results cannot be repeated or verified by an independent researcher.

Major issues

1. The data used for the analysis should be stored in a major database and preferably made publicly available

2. The software used for the transcriptome and GSE analysis is not presented. The software and any code used for the analysis should be available to the scientific community.

3. The potential targets for the pathways identified in the study should be part of the results section, instead of being incorporated in the discussion. A table would make the presentation of these results easier to read. 

4. The methodology for the signal pathways of the different cell types has been completely omitted. It should be presented in detail so that other researchers may be able to repeat it. 

5. The authors need to explain more vigorously the choice of 4 instead of 3 clusters in the pathway analysis since Figure 2D does not adequately support this choice.

Minor issues

1. Please include in the abstract the type of analysis performed (e.g. RNA seq gene expression)

2. The phrase in lines 43 to 46 feels like a repetition of the previous phrases

3. Please re-write the paragraph between lines 73 to 80 so that it clearly presents the aim of the study

4. line 89 please clarify the age criteria for eligible patients.

5. Please rewrite the phrases between lines 92 to 96 (repeats and missing verbs) 

6. line 103: replace "progressive" with "advanced"

7. What do MAD10/20/30/40 mean in Figure 1?

8. Figure 3A feels like a self-fulfilling prophecy since the clustered genes that popped up in the analysis from these patients were used to perform the clustered pathway analysis. Therefore, the agreement is not notable but expected.

Comments on the Quality of English Language

This manuscript would benefit significantly by being edited by someone with experience in scientific English.

Reviewer 3 Report

Comments and Suggestions for Authors

In the manuscript “Molecular clustering of metabolic dysfunction–associated steatotic liver disease (MASLD) based on transcriptome analysis”, the authors have described the validation of their MASLD cohort’s data into clusters to enable differential therapeutics to the patients with MASLD. The authors have presented the data in a constructive way. I have raised few comments during the evaluation of the manuscript. I suggest following comments to the authors to respond in the revised manuscript.

1.    In page 2 of 15, line 49-55, the paragraph explained about MASLD but, the reference 6 (Yang et al.) that you have mentioned was about triple negative breast cancer. I hope it should be 16th reference. Please clarify.

2.    I would like to know how it would be effective treatment to the patients by analyzing their specific gene signature and molecular marker of signaling mechanism, when compared to the systemic biomarkers of MASLD cohorts Which more useful for treating the patients with MASLD.  Required clarification.

3.    Mention no. of participants detail such as inclusion and exclusion criteria of the study in the materials and method. Possibly with a flow chart.

4.    I would like to suggest the authors to increase the font size of the Figure 2 A-F, 4A-F, and 5A. It is difficult to read and analyze the data presented.

5.    In page 8 of 15, line 256, it should be “no significant”. Check the typographical error.

6.    In discussion section, the authors need to expand the abbreviations that were being used such as PRKAG, TNF, BRD, MTR, CCR5, THRB, HMGCR, RAP, and more.

7.    Clustering the diversified MASLD cohort’s data seem to be interesting but, perhaps validation with other external study is necessary in order to confirm the therapeutic approach to the Patients. Please clarify this.

8.    The authors need to present detailed conclusion.

Round 2

Reviewer 1 Report

Comments and Suggestions for Authors

the authors have addressed all of my comments

Reviewer 2 Report

Comments and Suggestions for Authors

The authors have responded properly to my previous points